# Wavelet Flow: Fast Training of High Resolution Normalizing Flows

**Jason J. Yu**[1,3], **Konstantinos G. Derpanis**[2,4,5] **and Marcus A. Brubaker**[1,3,5]
[1]Department of Electrical Engineering and Computer Science, York University, Toronto
[2]Department of Computer Science, Ryerson University, Toronto
[3]Borealis AI, [4]Samsung AI Centre Toronto, [5]Vector Institute
jjyu@eecs.yorku.ca    kosta@cs.ryerson.ca    mab@eecs.yorku.ca

## Abstract

Normalizing flows are a class of probabilistic generative models which allow for both fast density computation and efficient sampling and are effective at modelling complex distributions like images. A drawback among current methods is their significant training cost, sometimes requiring months of GPU training time to achieve state-of-the-art results. This paper introduces *Wavelet Flow*, a multi-scale, normalizing flow architecture based on wavelets. A Wavelet Flow has an explicit representation of signal scale that inherently includes models of lower resolution signals and conditional generation of higher resolution signals, *i.e.*, super resolution. A major advantage of Wavelet Flow is the ability to construct generative models for high resolution data (*e.g.*, $1024 \times 1024$ images) that are impractical with previous models. Furthermore, Wavelet Flow is competitive with previous normalizing flows in terms of bits per dimension on standard (low resolution) benchmarks while being up to $15\times$ faster to train.

## 1   Introduction

Here we introduce *Wavelet Flow*, a multi-scale, conditional normalizing flow architecture based on wavelets. Wavelet Flows are not only fast when sampling and computing probability density, but are also efficient to train even with high resolution data. Further, the model has an explicit, wavelet-based, representation of scale which, among other benefits, includes consistent models of low resolution signals and conditional generation of higher resolutions. Wavelet Flow is applicable to any suitably structured data domain including audio, images, videos, and 3D scans. Our experiments focus on images as they are the most widely considered domain. The results demonstrate that Wavelet Flow is competitive with state-of-the-art normalizing flows on (typically low resolution) standard benchmarks while being significantly more efficient to train, *e.g.*, up to 15 times more in some cases. Further, we introduce the first normalizing flow model trained on high resolution images, *i.e.*, $1024 \times 1024$.

Normalizing flows are a class of probabilistic generative models which enable fast density computation and efficient sampling [9, 10, 26, 35]. They have also been shown to be effective at modelling complex distributions like images [24]. However, current approaches come with a significant computational cost, typically requiring months of GPU training time to achieve state-of-the-art results. This is partly driven by the requirement that normalizing flows be invertible and preserve dimensionality. Dinh et al. [10] identified this challenge and introduced a form of multi-scale flow which uses an identity transform on increasing subsets of dimensions to reduce computation. As an addition to this multi-scale flow, Ardizzone et al. [2] further proposed using a Haar wavelet transform for reshaping and reducing spatial resolution within a normalizing flow. In either case, the structure progressively limits the complexity of the transformation in certain dimensions but neither exposes nor exploits the natural scale structure in the signal itself. In contrast, Wavelet Flow uses a wavelet transformation

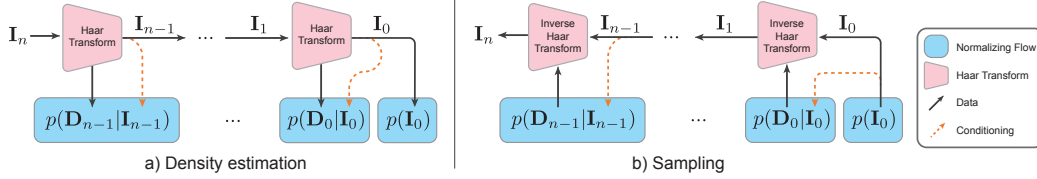

Figure 1: The architecture of a Wavelet Flow during (a) density estimation, and (b) sampling. The normalizing flows at each level (blue) can be trained independently from each other for easy parallelization. Sampling is performed in a coarse-to-fine manner, where each flow is conditioned (orange) on the lower resolution image.

[33] of the data that naturally exposes its inherent scale structure. This transformation enables a factorization of the distribution across scale and a novel conditional architecture, where coarse structure is generated first with the generation of increasingly fine details following, conditioned on the coarser structure.

The conditional structure of Wavelet Flow is reminiscent of autoregressive models [42, 40]. In these methods, conditioning is defined by an arbitrary pixel ordering, whereas the conditioning in Wavelet Flow is global but resolution limited. van den Oord et al. [42] also described a multi-scale variant of the PixelRNN that generates a full higher resolution image conditioned on a lower resolution input; however, this structure does not limit the conditional generation to the added content of the higher resolution image. Hence such a model can only be used for generation and does not allow for inference, *i.e.*, computing the probability density of a high resolution image. In contrast, Wavelet Flow conditionally generates only the detail coefficients of the wavelet representation at each scale and allows for inference of the probability density of an image at each modelled scale. Reed et al. [37] described a multi-scale autoregressive model, which consists primarily of an alternative ordering of the pixels rather than a direct representation of image scale.

In this work, we introduce the use of a multi-scale signal decomposition in normalizing flows. In general, multi-scale representations have been an area of significant interest in computer vision for decades [6, 34, 45, 5, 7, 27, 1, 32, 29, 30, 4]. Their use is driven by the inherent scale structure that exists in natural signals [29]. The Fourier basis [3] is perhaps the best known multi-scale decomposition and is both invertible and orthonormal; however, its global basis functions render modelling in the Fourier domain challenging. Gaussian and Laplacian pyramids [6, 5, 7, 1, 29, 30] provide an overcomplete but local representation for efficient coding and scale-sensitive representations. Mallat [32] showed that wavelets can be used as a multi-scale, orthogonal, and local representation of discrete signals, which have since been widely used for a variety of applications including compression and restoration. Here, we show how the orthogonality and spatial locality of wavelets make them uniquely well suited for use in normalizing flows.

Multi-scale image representations have also been explored in other forms of generative modelling, including generative adversarial networks (GANs) [12] and variational auto-encoders (VAEs) [25]. Denton et al. [8] proposed a hierarchical GAN consisting of a sequence of conditional GANs which generate the residual images of a Laplacian pyramid [5] representation. This is similar to our method which conditionally models the detail coefficients of each scale of a wavelet representation. However, the Laplacian pyramid is an overcomplete representation making it unsuitable for use in a normalizing flow. Various subsequent works [47, 21, 43, 20, 48, 41] also follow a stage-wise multi-scale image generation approach but forego the residual modelling and directly generate higher resolution images given lower resolutions inputs. PixelVAE [14] uses a hierarchical structure in their latent space but does not explicitly represent signal scale. Razavi et al. [36] extended the vector quantized variational autoencoder (VQ-VAE) model [42], by introducing scale dependent (discrete) latent codes, independently capturing local (*e.g.*, texture) and more global image aspects (*e.g.*, objects). Image generation is realized by a multi-stage PixelCNN [42] capturing the priors over the various scale-dependent latent maps. Dorta et al. [11] used a Laplacian pyramid with conditional VAEs in a manner similar to Wavelet Flow but primarily focus on image editing applications.

**Contributions** We propose the use of wavelets in normalizing flows. Orthonormal wavelets produce a multi-scale signal representation and, as we show, are convenient for use as part of a normalizing flow. We show that the distribution over a wavelet representation of a signal has a natural coarse-to-

fine conditional decomposition, which has a number of benefits including efficient and highly parallel training. Other benefits we explore include embedded, self-consistent distributions of lower resolution signals, and conditional generation of higher resolution signals (*i.e.*, super resolution). We also exploit this decomposition to introduce a novel sampling algorithm to draw samples from an annealed version of the resulting distribution. We apply the Wavelet Flow architecture to images, including $1024 \times 1024$ high resolution imagery which has not been previously modeled with normalizing flows. Compared against other normalizing flows on standard datasets, we demonstrate that Wavelet Flow has competitive performance while being up to $\sim 15\times$ faster to train. Code for Wavelet Flow is available at the following project page: `https://yorkucvil.github.io/Wavelet-Flow`.

## 2   Methods

In this section, the specifics of Wavelet Flow are described. To begin, wavelets are introduced (Sec. 2.1), then normalizing flows are briefly described (Sec. 2.2). Next, we describe how wavelets are used with normalizing flows to construct a Wavelet Flow (Sec. 2.3).

### 2.1   Wavelets

Wavelets are a multi-scale decomposition of a signal similar in concept to a Fourier basis representation. While a Fourier basis is global in nature, wavelets are constructed to be localized, meaning that the value of a wavelet coefficient reflects the structure of the signal in a local region. This is particularly beneficial as modern deep learning architectures generally, including normalizing flows, are well tailored to spatially structured signal representations due to the widespread use of convolutional operations. Here, we briefly introduce wavelets; see [33] for a thorough introduction.

Consider an image $\mathbf{I} \in \mathbb{R}^{2^n \times 2^n \times C}$. The discrete wavelet transformation is constructed recursively as

$$\mathbf{I}_{i-1} = h_l(\mathbf{I}_i) \quad \text{and} \quad \mathbf{D}_{i-1} = h_d(\mathbf{I}_i), \tag{1}$$

where $\mathbf{D}_{i-1} \in \mathbb{R}^{2^{i-1} \times 2^{i-1} \times 3C}$ are the detail coefficients at level $i-1$, $h_l(\mathbf{I})$ applies the wavelet's low-pass filter (channel-wise), $h_d(\mathbf{I})$ applies the wavelet's high-pass filters (channel-wise), and both $h_l(\mathbf{I})$ and $h_d(\mathbf{I})$ use a stride of two, *i.e.*, spatial downsampling the result by a factor of two. The wavelet transform is invertible and the original image can be recovered recursively by

$$\mathbf{I}_i = h^{-1}(\mathbf{I}_{i-1}, \mathbf{D}_{i-1}), \tag{2}$$

where $h^{-1}$ is the inverse wavelet transform [33]. Applying the wavelet transform recursively until level 0 yields a complete representation of the image, where $\mathcal{H}(\mathbf{I}) = (\mathbf{I}_0, \mathbf{D}_0, \mathbf{D}_1, \mathbf{D}_2, \ldots, \mathbf{D}_{n-1}) \in \mathbb{R}^{C2^n}$ denotes the full wavelet transformation of an image. Note that $\mathcal{H}(\mathbf{I})$ has the same total dimensionality as $\mathbf{I}$. This representation decomposes the content of the image into a range of scales while retaining the dimensionality. At the coarsest scale, $\mathbf{I}_0 \in \mathbb{R}^{1 \times 1 \times C}$ represents the average intensity per channel and $\mathbf{D}_0 \in \mathbb{R}^{1 \times 1 \times 3C}$ represents the global variations. At the finest scale, $\mathbf{D}_{n-1} \in \mathbb{R}^{2^{n-1} \times 2^{n-1} \times 3C}$ represents local variations.

There are many wavelets which have been designed and could be applied here. For Wavelet Flow we choose to use an orthonormal wavelet as described below. For simplicity, a Haar wavelet [15, 33] is used; however, any orthonormal wavelet could be used. Haar wavelets are fast and simple to apply and use only $2 \times 2$ filters. Additional details for the Haar transformation are available in the Supplemental Material. Exploration of other wavelets remains a direction for future work.

### 2.2   Normalizing Flows

Normalizing Flows are an application of the change of variables formula for probability density estimation. We briefly introduce the core elements here, but for a full review, see [26, 35]. If $\mathbf{X}$ is a random variable with values $\mathbf{x} \in \mathbb{R}^D$ then the density function of $\mathbf{X}$ can be written as

$$p_{\mathbf{X}}(\mathbf{x}) = p_{\mathbf{Z}}(\mathbf{f}_\theta(\mathbf{x})) |\det \mathbf{J}_\theta(\mathbf{x})|, \tag{3}$$

where $\mathbf{Z}$ is a random variable which takes on values $\mathbf{z} \in \mathbb{R}^D$, with probability density function $p_{\mathbf{Z}}(\mathbf{z})$, and with a known distribution which we will assume to be Normal with mean 0 and unit variance. $\mathbf{f}_\theta$ is an invertible, differentiable function parameterized by $\theta$. $\mathbf{J}_\theta = \frac{\partial \mathbf{f}_\theta}{\partial \mathbf{x}}$ is the Jacobian of $\mathbf{f}_\theta$. Learning

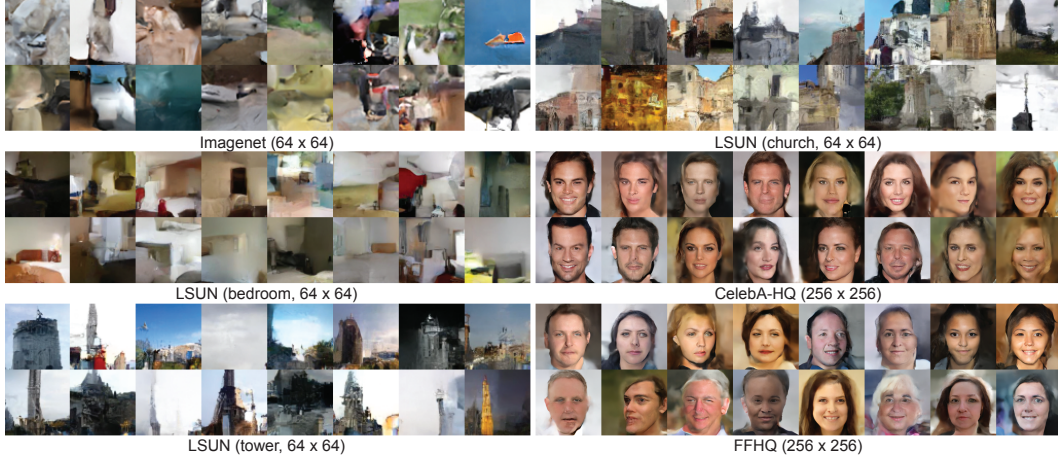

Imagenet (64 x 64)            LSUN (church, 64 x 64)

LSUN (bedroom, 64 x 64)            CelebA-HQ (256 x 256)

LSUN (tower, 64 x 64)            FFHQ (256 x 256)

Figure 2: Wavelet Flow samples drawn using MCMC with $T = 0.97$.

in normalizing flows consists of optimizing the parameters $\theta$ to maximize the log likelihood of a set of observations. The resulting distribution can be easily sampled from by drawing samples $\mathbf{z} \sim p_{\mathbf{Z}}(\mathbf{z})$ and then applying the inverse transformation to yield $\mathbf{x} = \mathbf{f}_{\theta}^{-1}(\mathbf{z})$.

The core challenge is constructing invertible and differentiable functions which are also efficient to compute and have easily computable Jacobian determinants. Two widely used functions are *affine* [10] and *additive* [9] *coupling* layers. Affine coupling layers split the input into two disjoint subsets, $\mathbf{x} = (\mathbf{x}^A, \mathbf{x}^B)$, and applies the transformation $\mathbf{f}_{\theta}(\mathbf{x}) = (\mathbf{x}^A, \mathbf{s}_{\theta}(\mathbf{x}^A) \odot \mathbf{x}^B + \mathbf{t}_{\theta}(\mathbf{x}^A))$, where $\odot$ denotes element-wise multiplication, and $\mathbf{s}_{\theta}(\mathbf{x}^A)$ and $\mathbf{t}_{\theta}(\mathbf{x}^A)$ are arbitrary functions, typically deep networks of some form, called *coupling networks*. The Jacobian determinant of an affine coupling layer is simply the product of the values of $\mathbf{s}_{\theta}(\mathbf{x}^A)$, and its inverse is straightforward. An additive coupling layer is simply an affine coupling layer without the scaling operation, *i.e.*, $\mathbf{f}_{\theta}(\mathbf{x}) = (\mathbf{x}^A, \mathbf{x}^B + \mathbf{t}_{\theta}(\mathbf{x}^A))$, and has unit Jacobian determinant.

## 2.3 Wavelet Flow

To derive Wavelet Flow, we apply the change of variables to arrive at a distribution of the wavelet coefficients. Specifically, $p(\mathbf{I}) = p(\mathcal{H}(\mathbf{I}))|\det \frac{\partial \mathcal{H}}{\partial \mathbf{I}}|$, where we drop the random variable subscripts for clarity. To easily calculate the determinant of the transform, we require that the selected wavelet is orthonormal[1] and hence $|\det \frac{\partial \mathcal{H}}{\partial \mathbf{I}}| = 1$. One can now apply the product rule of probability to conditionally factorize the distribution, giving

$$p(\mathbf{I}_n) = p(\mathbf{I}_0) \prod_{i=0}^{n-1} p(\mathbf{D}_i | \mathbf{I}_i), \tag{4}$$

where each $\mathbf{I}_i$ is constructed using the inverse wavelet transform. In other words, instead of modelling the entire image directly, a Wavelet Flow models the distribution of detail coefficients, conditioned on the lower resolution image. Each conditional distribution of detail coefficients, $p(\mathbf{D}_i | \mathbf{I}_i)$, and distribution of average intensity, $p(\mathbf{I}_0)$, is constructed using a normalizing flow. Note that inherent in this structure are distributions of all coarser resolution images, *e.g.*, $p(\mathbf{I}_k) = p(\mathbf{I}_0) \prod_{i=0}^{k-1} p(\mathbf{D}_i | \mathbf{I}_i)$ for all $k \leq n$. Hence, a Wavelet Flow model trained at a higher resolution can be used as a model of lower resolution images. The general structure of a Wavelet Flow is shown in Fig. 1 and the specific normalizing flow architecture used for these distributions is described in Sec. 3.

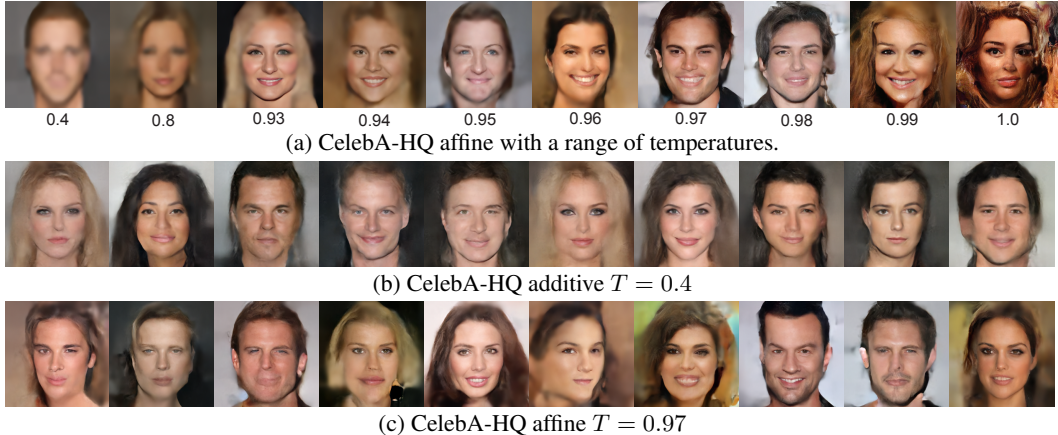

(a) CelebA-HQ affine with a range of temperatures.

(b) CelebA-HQ additive $T = 0.4$

(c) CelebA-HQ affine $T = 0.97$

Figure 3: Annealed samples from additive and affine Wavelet Flows on CelebA-HQ.

**Training**  Similar to other normalizing flow models, a Wavelet Flow is trained by maximizing the log likelihood. Specifically, we seek to maximize the sum of

$$\log p(\mathbf{I}) = \log p(\mathbf{I}_0) + \sum_{i=0}^{n-1} \log p(\mathbf{D}_i|\mathbf{I}_i) \tag{5}$$

over a set of sample images. This reveals a significant advantage of a Wavelet Flow: the conditional distribution of detail coefficients for each level can be trained *independently*. This makes training more efficient, as it is easily parallelized with no communication overhead, gradient checkpointing or approximations, and the individual distributions are generally smaller and simpler, *i.e.*, they describe lower-dimensional distributions making them easier to fit in limited GPU memory.

**Sampling**  Sampling from a Wavelet Flow starts with $\mathbf{I}_0 \sim p(\mathbf{I}_0)$ and proceeds recursively with

$$\mathbf{D}_{i-1} \sim p(\mathbf{D}_{i-1}|\mathbf{I}_{i-1}) \text{ and } \mathbf{I}_i = h^{-1}(\mathbf{I}_{i-1}, \mathbf{D}_{i-1}), \tag{6}$$

where sampling from each distribution is simply sampling from the corresponding normalizing flow, *i.e.*, sampling from the base distribution of that flow and applying its inverse flow transformation. However qualitative results in normalizing flow papers (*e.g.*, [24, 31]) are typically produced from an annealed distribution $\propto p(\mathbf{I})^{1/T^2}$ with $T < 1$ a temperature parameter. Annealing tightens the distribution around its modes and reduces the amount of spurious samples along with limiting sample diversity. If the normalizing flow has a constant Jacobian determinant term (*e.g.*, it consists only of additive coupling layers) then $p(\mathbf{I})^{1/T^2} \propto p_{\mathbf{Z}}(\mathbf{f}(\mathbf{I}))^{1/T^2}$ and hence samples can be drawn from a Gaussian distribution with its standard deviation scaled by $T$ and applying the inverse of the flow transformation. This approach does not work with more general flows including, *e.g.*, the affine coupling flows used here. Consequently, previous approaches resort to restrictive models with additive couplings when annealing the distribution for qualitative evaluation while using more expressive models with affine couplings for quantitative evaluation. In this work, we opt to use a single model with affine couplings for both quantitative and qualitative evaluation.

To sample from an annealed flow with affine couplings, we apply Markov Chain Monte Carlo (MCMC) to the annealed, unnormalized distribution $p(\mathbf{I})^{1/T^2}$. Specifically, we use the No-U-Turn Sampler (NUTS) algorithm [17, 28] to generate samples. However, the high dimensionality of images can make this expensive. A Wavelet Flow allows us to accelerate sampling by running MCMC separately at each scale and using the resulting samples to condition the next scale. This enables a more efficient sampling due to the lower dimensionality of each scale distribution. For full details of the MCMC approach, see the Supplemental Material.

## 3  Experimental Evaluation

As described in Section 2, the distributions $p(\mathbf{D}_i|\mathbf{I}_i)$ and $p(\mathbf{I}_0)$ are constructed using normalizing flows. Specifically, a Glow architecture [24] is used with slight modifications. Since wavelets are

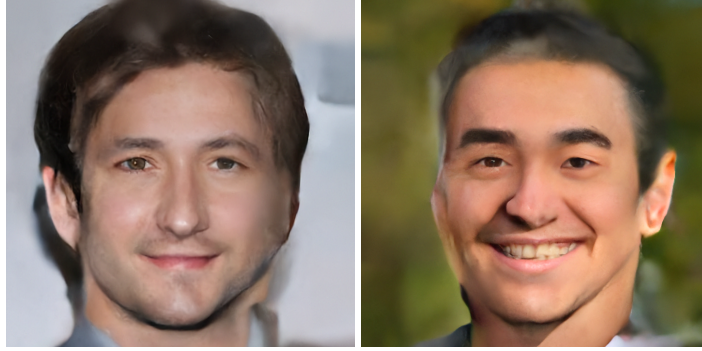

Figure 4: Wavelet Flow samples at $1024 \times 1024$ on CelebA-HQ (left) and FFHQ (right) with $T = 0.97$.

used to decompose the image at multiple scales, the multi-scale squeeze/split architecture from [10] is not used. For the conditional flows, the lower resolution image is concatenated to the input of each coupling network. Learning a conditional transformation of base distributions [44] was explored but not found to improve performance. The coupling networks use a residual architecture with a final zero initialized convolutional layer as in [24].

Training is done using the same Adamax optimizer [23] as in [24]. The architecture of the flow is fully convolutional, and patch-wise training is used for the highest resolution conditional distributions. In cases where overfitting is observed, early stopping is applied based on a held-out validation set. These implementation choices allow for distributions to be trained using a batch-size of 64 without gradient checkpointing on a single NVIDIA TITAN X (Pascal) GPU.

For evaluation baselines, we consider RealNVP [10] and Glow [24] with uniform dequantization for all approaches. Comparison against these methods is only possible on lower resolution datasets as training on high resolution datasets is impractical. We note that improved architectures for a Wavelet Flow remains a direction of future work and could exploit newer flow architectures (*e.g.*, MaCow [31], Flow++ [16], and SoS Flow [19]) and alternative dequantization schemes [16, 18]. Hyper-parameters are set to produce models with parameter counts similar to but not exceeding those of Glow [24] to enable a fair comparison. This requires per-dataset tuning as other approaches used dataset specific architectures. Parameter counts and specific hyper-parameter choices can be found in the Supplemental Material.

**Datasets and Evaluation**    To evaluate the performance of Wavelet Flow, we use several standard image datasets to directly compare against the reported results of previous methods. Specifically, we train and evaluate our model on natural image datasets at the commonly used resolutions and follow standard preprocessing: *ImageNet* [38] ($32 \times 32$ and $64 \times 64$) and *Large-scale Scene Understanding (LSUN) bedroom*, *tower*, and *church* outdoor [46] ($64 \times 64$). We also train on two high resolution datasets at resolutions not previously reported: *CelebFaces Attributes High-Quality (CelebA-HQ)* [21] ($1024 \times 1024$) and *Flickr-Faces-HQ (FFHQ)* [22] ($1024 \times 1024$). We use the same CelebA-HQ dataset split used in [24]. FFHQ contains 70 000 high quality $1024 \times 1024$ images of faces aligned and cropped from Flickr. This high resolution dataset has not been previously used in conjunction with normalizing flows. As no standard dataset split is available, we generate our own with 59 000, 4 000, and 7 000 images for training, validation and testing, respectively. Quantitative evaluations are based on average log likelihoods over a test set in the form of the commonly reported *bits per dimensions* (BPD) corresponding to intensity discretization into 256 bins. All experiments are performed with the original 8-bit images unless otherwise noted. Kingma and Dhariwal [24] reported that reducing the bit-depth of the data to 5-bit can improve the quality of samples; however, no such effect was observed with Wavelet Flow and consequently we use the original bit-depth of the data. For full details of the datasets, their experimental setup, and 5-bit quantitative results, see the Supplemental Material.

**Results**    Quantitative results in BPD are shown in Table 1. The Wavelet Flow models are competitive with other methods, outperforming Glow on ImageNet while somewhat underperforming Glow on LSUN but with significantly faster training. Training times for each dataset and scale vary with the

| Model | ImageNet [38] | | LSUN [46] bedroom tower church | | | CelebA-HQ [21] | FFHQ [22] |
|---|---|---|---|---|---|---|---|
| | $32 \times 32$ | $64 \times 64$ | $64 \times 64$ | | | $1024 \times 1024$ | $1024 \times 1024$ |
| RealNVP [10] | 4.28 | 3.98 | 2.72 | 2.81 | 3.08 | - | - |
| Glow [24] | 4.09 | 3.81 | 2.38 | 2.46 | 2.67 | - | - |
| Wavelet Flow | 4.08 | 3.78 | 2.41 | 2.49 | 2.74 | 1.34 | 2.07 |

Table 1: Quantitative performance in bits per dimension. RealNVP and Glow are not evaluated on CelebA-HQ 1024 or FFHQ 1024 as training is impractical at those resolutions.

smallest scales taking a few hours and largest scales typically requiring five or six days. The total training times for ImageNet at $64 \times 64$ and CelebA-HQ at $1024 \times 1024$ were 822 and 672 GPU hours, respectively. Exact training times for Glow [24] were not reported in the paper but are estimated[2] to be approximately $6,700$ GPU hours to train on CelebA-HQ at $256 \times 256$ and $3,700$ GPU hours to train on LSUN bedroom at $64 \times 64$. In contrast, training Wavelet Flow on those datasets and resolutions take approximately 430 and 572 GPU hours, respectively, $15\times$ and $6\times$ faster than Glow. We also measure the average number of seconds-per-image over 100 iterations of training on our hardware. On CelebA-HQ at $256 \times 256$, ImageNet at $64 \times 64$, and LSUN at $64 \times 64$, we achieve speedups greater than $30\times$ that of Glow, in terms of seconds-per-image. This is enabled, in part, because the smaller, simpler individual conditional models that make up Wavelet Flow allow for larger batch sizes than is possible with a single monolithic Glow model and their training can be more efficiently parallelized. More detailed training times can be found in the Supplemental Material.

As described above, a Wavelet Flow can automatically be applied on lower resolution signals. To demonstrate this we use the model trained on ImageNet at $64 \times 64$ and truncate it to evaluate at $32 \times 32$, getting a BPD of 4.16 vs. 4.08 with the model trained directly at $32 \times 32$. This difference is due to dequantization [16, 18]. Specifically, dequantization noise is low-pass filtered in the wavelet transform along with the original signal and is no longer uniform at lower resolutions and hence there are small (below a single pixel intensity increment) differences in the data distribution for the embedded lower resolution models. This was verified by using dequantization noise consistent with this low-pass filtered uniform noise during testing of the truncated model which achieves a BPD of 4.08, exactly the same as a Wavelet Flow trained directly on $32 \times 32$ images. Note that this does not occur when modelling a truly continuous signal, where dequantization is not necessary. Further exploration of dequantization in Wavelet Flow remains a direction for future work.

To assess the qualitative performance of our model, we sampled from an annealed version of the estimated distribution using the MCMC approach described in Section 2.3. We explore a range of annealing parameters and find that only a small amount of annealing, *i.e.*, $T = 0.97$, is required to produce good visual results. In contrast, other methods typically require a large amount of annealing to produce plausible images, *e.g.*, Glow [24] reported results using $T = 0.7$ for CelebA-HQ. Generated images for all datasets are shown in Fig. 2 and more are in the Supplemental Material. Images generated at different values of $T$ can be found in Fig. 3(a). Notably, at extremely low temperatures the images become somewhat blurry, likely indicating that the annealed distribution of detail coefficients (see below) has become too peaked at zero at finer scales. Further, lower values of $T$ may cause problems with convergence of the MCMC based sampling approach.

High resolution samples from Wavelet Flow trained on CelebA-HQ and FFHQ at $1024 \times 1024$ are shown in Figure 4 and more are in the Supplemental Material. We also report a quantitative evaluation of Wavelet Flow on these datasets in Table 1 but note that comparison to other methods is not possible as training the baselines on high resolution imagery is impractical.

**Super Resolution** The conditional structure of a Wavelet Flow allows them to perform probabilistic super resolution by generating the detail coefficients given a lower resolution input image. That is, given an input image as $\mathbf{I}_i$ we can generate $\mathbf{D}_i$ by sampling from $p(\mathbf{D}_i|\mathbf{I}_i)$ and construct $\mathbf{I}_{i+1} = h^{-1}(\mathbf{I}_i, \mathbf{D}_i)$ to achieve a $2\times$ increase in resolution. This process can be applied iteratively to recover higher resolutions. An example of $8\times$ upsampling, from $128 \times 128$ to $1024 \times 1024$, can be seen in Fig. 5. Note that our model is not trained specifically for super resolution, rather this is a notable byproduct of the scale conditional structure of Wavelet Flow. We present these results only as a demonstration and leave a full exploration of Wavelet Flow for super resolution as future work.

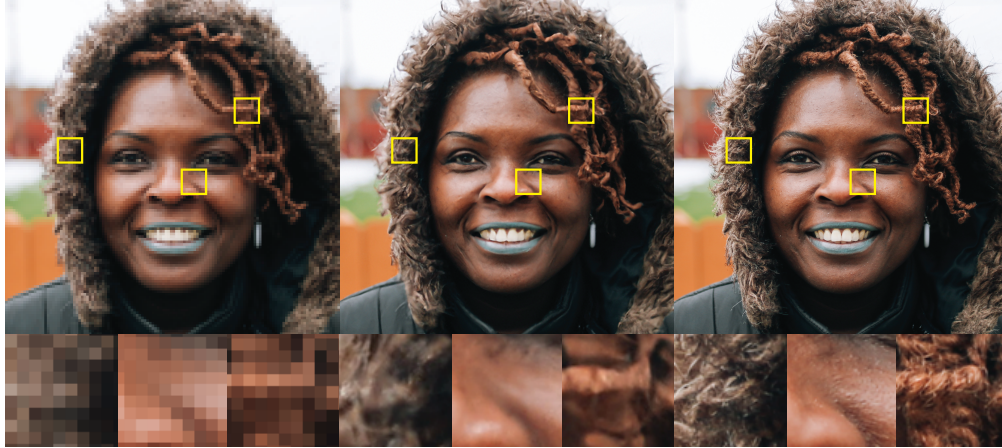

Figure 5: Wavelet Flow super resolution with $T = 0.97$ trained on FFHQ from $128 \times 128$ (left) to $1024 \times 1024$ (middle) with ground truth (right) and magnified regions (bottom) for comparison.

**Effects of Annealing** Training with maximum likelihood (ML) typically results in models that favour diversity rather than specificity due to its "mean-seeking" tendency. Kingma and Dhariwal [24] further speculated that flow-based models over-estimate the entropy of the distribution. As a result, images directly drawn from ML trained models tend to be less realistic. Previous flow-based approaches (*e.g.*, [24, 31]) have instead shown samples drawn from an annealed version of the estimated distribution. We can see this over estimation of the entropy in the marginal distribution of the generated detail coefficients shown in Fig. 6. Specifically, without annealing the peak of the learned distribution of wavelet details is lower. However, after annealing, the distribution of detail coefficients matches well with the data, further supporting the choice of

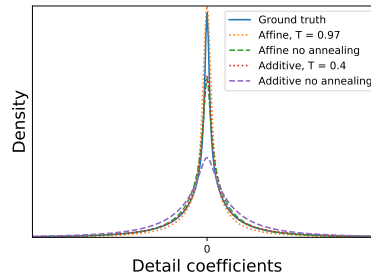

Figure 6: Marginal distributions of detail coefficients $\mathbf{D}_5$ for Wavelet Flow trained on CelebA-HQ.

$T = 0.97$. In contrast, with the additive model we see that even with $T = 0.4$ the distribution still does not match.

**Additive vs. Affine Coupling** As discussed in Section 2.3, to draw samples from an annealed distribution in previous works (*e.g.*, [24, 31]), affine coupling layers are replaced with additive coupling layers to produce a model with a constant Jacobian determinant. While affine layers generally produce better quantitative results [10, 24], it is unknown whether they are also qualitatively better when annealed. Here, we present the first direct comparison between annealed additive and affine models. Fig. 3(b) and 3(c) compare samples from Wavelet Flows with additive and affine layers. Our results demonstrate that, in the case of Wavelet Flow, additive and affine layers have similar image quality. However, more aggressive annealing of the additive model is required. Further, as shown in Fig. 6, even with aggressive annealing, additive models still cannot match the marginal statistics of the detail coefficients as well as affine models. As a result we choose only to use a single, affine coupling-based model for both quantitative evaluation and sample generation.

**Evaluation of Sample Quality** Qualitatively Wavelet Flow appears to have somewhat worse performance than the Glow baseline. This is corroborated by FID scores [39] provided in the Supplemental Material, but can also be seen in one specific aspect: spatially distant dependencies are not well captured by Wavelet Flow. In particular, global coherence of fine details, *e.g.*, eye colour, gaze direction, and hair texture, appear inconsistent over larger distances. This is most obvious when modelling human faces, and manifests as asymmetric eyes shown in Fig. 7, and inconsistent hair texture shown in Fig. 8. It is not immediately clear exactly how the wavelet decomposition, receptive field, patch based training, and conditioning, interact to cause this. Analysing the detail coefficients in Fig. 8 suggests that these inconsistencies mainly occur at higher frequencies.

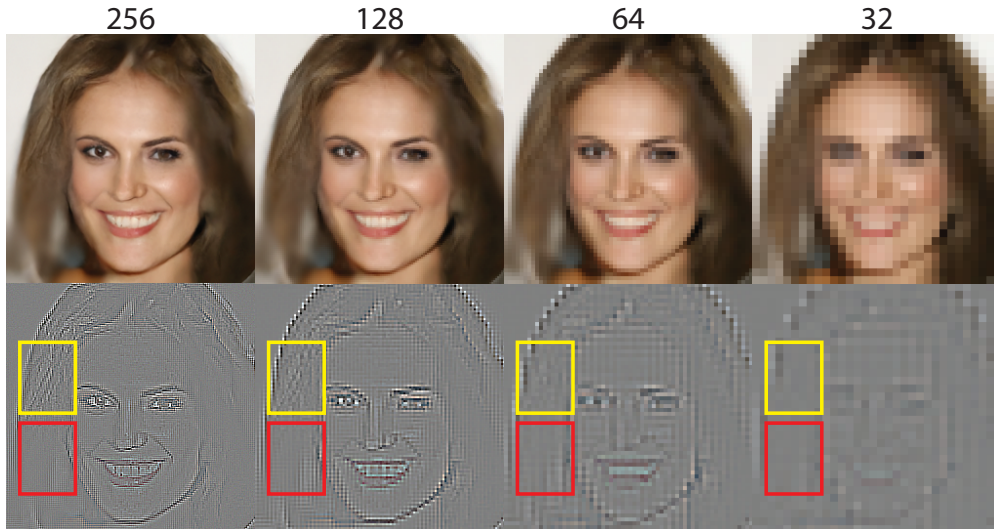

Figure 8: Texture inconsistency over large distances with Wavelet Flow. A sample from Wavelet Flow trained on CelebA-HQ at different scales (top), and visualizations of the corresponding detail coefficients (bottom). Two regions of interest are highlighted by yellow and red boxes. The yellow region has hair texture, while the red region is overly smooth. The corresponding regions in the detail coefficient visualizations show that the inconsistencies are mainly due to differences in the high frequencies details.

## 4 Discussion and Future Work

In this paper, we introduced Wavelet Flow, a multi-scale, conditional normalizing flow architecture based on wavelets. We showed that the resulting model is up to $15\times$ faster to train and enables the training of generative models for high resolution data which is impractical with previous flow-based approaches. We also showed how the multi-scale structure of our Wavelet Flow model could be used to extract consistent distributions of low resolution signals, as well as to perform super resolution.



There remains several promising directions for future work. The choice of wavelet was not considered here; a Haar wavelet was used for simplicity. Numerous wavelets exist [33] and it is unclear how the choice of wavelet may impact performance. This paper focused on natural images, but Wavelet Flow is directly applicable to other domains, *e.g.*, video, medical imaging, and audio. With images there remains work to explore the use of Wavelet Flow for problems such as image restoration and super resolution. While we explored some architecture choices for the conditional flows, we expect that improvements in performance will be found with other architectures, *e.g.*, [16, 31, 13, 19]. Initial experiments investigating the reduced global coherence in Wavelet Flow were inconclusive; however, we suspect that further architecture search may help address

Figure 7: Detail inconsistency over large distances with Wavelet Flow. A sample from Wavelet Flow trained on CelebA-HQ at different resolutions enlarged around the asymmetrically coloured eyes. The asymmetry only becomes apparent at higher resolutions.

these issues. Further, as noted in Sec. 3, dequantization in a Wavelet Flow can interfere with certain desirable multi-scale properties. Adapting dequantization for use in a Wavelet Flow may rectify this and is likely to improve performance as it has with other normalizing flow models [16, 18].

## Acknowledgments and Disclosure of Funding

This work was started as part of J.J.Y.'s internship at Borealis AI and was supported by the Mitacs Accelerate Program, funded in part by the Canada First Research Excellence Fund (CFREF) for the Vision: Science to Applications (VISTA) program (M.A.B.) and the NSERC Discovery Grant program (M.A.B., K.G.D.). K.G.D. contributed to this work in his capacity as an Associate Professor at Ryerson University.

## Broader Impact

This work contributes to the domain of generative image modelling, which constructs probabilistic models of images, allowing the generation, and manipulation of high resolution images. As this domain grows, these tasks become more accessible due to reduced labour, and skill barriers. On a positive side, this accessibility can empower artists and content creators with powerful tools for their craft. Many other processes used today are data driven, *e.g.*, semantic segmentation, and could benefit from the ability to create new data. Generative models are also useful in applications beyond content generation. In particular, they can be used as priors in other signal processing tasks including image restoration, and super resolution. The computational costs of machine learning are of increasing concern as a potential future source of social and economic inequality. Reducing the need for specialized, large-scale hardware for training, as the approach proposed here does, can lower the barrier to entry for individuals and groups which would otherwise be unable to afford the large compute clusters necessary.

As with all tools, generative models have no intentions of their own. Their impact is defined by those who wield them. Media platforms play an important role in our lives, and as a result can be used for disinformation. A major difficulty in preventing disinformation is the asymmetry between the capacity to create and detect disinformation. If the detection of disinformation is more difficult than its creation, its prevention may become impossible in practice. Our contributions primarily serve to reduce the costs of content generation. Specifically, the ability to operate on high resolution images at a reduced cost makes it easier for actors that wish to create convincing content in large volumes. Consequently, this could worsen the battle against disinformation.

Even without actors that are intentionally malicious, reduced barriers to these tools could increase the absolute amount of unintentional misuse. The nature of generative models encourages them to adopt biases in the data used to train them. Without caution, a user could introduce biases to a generative model that may discriminate against certain groups of people. This model could then be easily used to create large quantities of data that contain this bias. This could take the form of data used in other downstream processes, or directly as content consumed by people.

These concerns assume unopposed misuse of generative methods. Complementary research into the detection of generated content could mitigate these potential consequences. Much like how active research into adversarial attacks continuously competes with research into robust recognition methods, generative methods can exist at an equilibrium with methods used to detect their use, and biases, preventing runaway consequences.

## Footnotes

[1] As commonly implemented, the Haar wavelet filters are often unnormalized and hence orthogonal but not orthonormal. This can be rectified by a suitable rescaling.

[2]Training times for Glow were estimated by inspecting the available training logs and along with additional details reported by the lead author here: `https://github.com/openai/glow/issues/37`

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
