[Supplementary Material]

# Supplemental Material for *Wavelet Flow: Fast Training of High Resolution Normalizing Flows*

**Jason J. Yu**[1,3]**, Konstantinos G. Derpanis**[2,4,5] **and Marcus A. Brubaker**[1,3,5]
[1]Department of Electrical Engineering and Computer Science, York University, Toronto
[2]Department of Computer Science, Ryerson University, Toronto
[3]Borealis AI, [4]Samsung AI Centre Toronto, [5]Vector Institute
jjyu@eecs.yorku.ca   kosta@cs.ryerson.ca   mab@eecs.yorku.ca

## A   Supplementary code

Code and project page is available at: `https://yorkucvil.github.io/Wavelet-Flow`

## B   MCMC Sampling with Wavelet Flow

As discussed in Sec. 2.3, MCMC (*i.e.*, No-U-Turn Sampler (NUTS) algorithm [3, 7]) is used to draw annealed samples from the annealed Wavelet Flow model. In this section, we describe the annealed sampling process. First, we describe how MCMC can be used to draw samples from any distribution constructed as a normalizing flow. Next, we describe how the Wavelet Flow structure in particular is used to enable faster sampling.

### B.1   MCMC on an Annealed Flow

The target distribution for MCMC is the annealed normalizing flow and can be written as:

$$\pi_{\mathbf{X}}(\mathbf{x}) \propto p_{\mathbf{X}}(\mathbf{x})^{\gamma} \tag{1}$$
$$= p_{\mathbf{Z}}(\mathbf{f}(\mathbf{x}))^{\gamma} |\det \mathbf{J}(\mathbf{x})|^{\gamma}, \tag{2}$$

where $\gamma = 1/T^2$ is the annealing parameter, $\mathbf{f}$ is the normalizing flow transformation, $\mathbf{J}$ is the Jacobian of $\mathbf{f}$, and the degree of annealing is specified as a *temperature* with $T = 1/\sqrt{\gamma}$, following the convention of [6].

The No-U-Turn Sampler (NUTS) [3] requires only that the unnormalized probability density and its gradients can be evaluated and can be applied directly to $\pi_{\mathbf{X}}(\mathbf{x})$; however, the complex dependencies that exist between dimensions of $\mathbf{x}$ can produce a challenging geometry of the probability density which can be difficult for MCMC algorithms to sample from efficiently. Since we know the form of the density is closely related to a known normalizing flow, we can use the inverse of this flow, $\mathbf{g}$, to reparameterize the density such that it becomes exactly Gaussian (and hence easier to sample) when $\gamma = 1$. For $\gamma \neq 1$ the geometry should still be close to Gaussian and hence easier to sample from, particularly with values of $\gamma$ close to 1. Reparameterizing the annealed distribution in terms of $\mathbf{z}$ gives:

$$\pi_{\mathbf{Z}}(\mathbf{z}) \propto \pi_{\mathbf{X}}(\mathbf{g}(\mathbf{z}))|\det \mathbf{H}(\mathbf{z})| \tag{3}$$
$$= p_{\mathbf{Z}}(\mathbf{f}(\mathbf{g}(\mathbf{z})))^{\gamma} |\det \mathbf{J}(\mathbf{g}(\mathbf{z}))|^{\gamma} |\det \mathbf{H}(\mathbf{z})| \tag{4}$$
$$= p_{\mathbf{Z}}(\mathbf{z})^{\gamma} |\det \mathbf{H}(\mathbf{z})|^{1-\gamma}, \tag{5}$$

where $\mathbf{H} = \frac{\partial \mathbf{g}}{\partial \mathbf{z}}$ is the Jacobian of $\mathbf{g}$, and the last line follows because $\mathbf{g}$ is the inverse of the flow transformation $\mathbf{f}$, *i.e.*, $\mathbf{g} = \mathbf{f}^{-1}$, and hence their Jacobians (and their determinants) are also inverses of each other. NUTS is then used to perform MCMC on $\pi_{\mathbf{Z}}(\mathbf{z})$. Samples from this Markov chain

can then be transformed into samples from $\pi_{\mathbf{X}}(\mathbf{x})$ by computing $\mathbf{g}(\mathbf{z})$ like in a normalizing flow. In practice, we found that sampling in terms of $\mathbf{z}$ using the NUTS algorithm [3] is more efficient and can be done with a larger step size and fewer divergences, compared to sampling in terms of $\mathbf{x}$. We use the implementation of NUTS provided in [7].

### B.2 Multi-scale MCMC with Wavelet Flow

The above procedure can be inefficient if applied to the entire distribution of wavelet coefficients due to the high dimensionality and complexity of the distributions. Instead, it is applied levelwise. In particular, first samples are drawn using MCMC as described above from $p(\mathbf{I}_0)^\gamma$, then using the last sample from that chain we sample from $p(\mathbf{D}_0|\mathbf{I}_0)^\gamma$, construct $\mathbf{I}_1 = h^{-1}(\mathbf{I}_0, \mathbf{D}_0)$ and continue this procedure for $\mathbf{D}_1$, $\mathbf{D}_2$ and so on until all detail coefficients have been generated and a sample of $\mathbf{I}$ is drawn from the annealed distribution. This algorithm is shown in Algorithm S1.

---

**Algorithm S1:** Annealed sampling from Wavelet Flow using NUTS. $\gamma$ is the annealing parameter, $N$ is the number of levels in the Wavelet Flow, and $m$ is the minimum number of NUTS steps. In our experiments, $m$ is set to 30. $\mathbf{z}_i$ is the base value of the flow $\mathbf{g}_i$ used to produce $\mathbf{D}_i$. Subscript $b$ indicates components for the initial base image.

---

Given $\gamma, N, m$:
Sample $\hat{\mathbf{z}}_b \sim p_{\mathbf{Z}_b}^\gamma(\mathbf{z}_b)$
Sample $\mathbf{z}_b \sim \pi_{\mathbf{Z}_n}(\mathbf{z}_b)$ using NUTS, initialized with $\hat{\mathbf{z}}_b$, and $m$ steps
Set $\mathbf{I}_0 \leftarrow \mathbf{g}_b(\mathbf{z}_b)$
**for** $i \leftarrow 0$ **to** $N-1$ **do**
    Sample $\hat{\mathbf{z}}_i \sim p_{\mathbf{Z}_i}^\gamma(\mathbf{z}_i)$
    Sample $\mathbf{z}_i \sim \pi_{\mathbf{Z}_i}(\mathbf{z}_i|\mathbf{I}_i)$ using NUTS, initialized with $\hat{\mathbf{z}}_i$, and $m$ steps
    Set $\mathbf{D}_i \leftarrow \mathbf{g}_i(\mathbf{z}_i|\mathbf{I}_i)$
    Set $\mathbf{I}_{i+1} \leftarrow h^{-1}(\mathbf{I}_i, \mathbf{D}_i)$
**end**

---

In our experiments, the Markov chains are initialized with a sample from $p_{\mathbf{Z}}(\mathbf{f}(\mathbf{g}(\mathbf{z})))^\gamma$, which is straightforward since it is an annealed Gaussian and annealing simply scales the standard deviation. This provides a reasonable initialization which is exact in the case where the flow has a constant volume correction term or $\gamma = 1$. Samples from this initialization are not proper samples from the annealed distribution but can still look reasonable, further suggesting that it is a good initialization. Dual averaging adaptation is used to adjust the step size for the first 10 steps of NUTS [3]. A minimum of 30 steps are taken before the next accepted proposal is taken as the sample. These hyper-parameters were chosen since they gave good qualitative performance without using too much compute. In our experiments, we observed that it takes roughly 30 minutes to sample an annealed image at $256 \times 256$ using $T = 0.97$ on CelebA-HQ. It roughly takes an additional 50 minutes and 35 hours to further reach resolutions of $512 \times 512$ and $1024 \times 1024$, respectively. Levelwise sampling can exploit parallelism at coarser scales but GPU memory limits when sampling details at higher resolutions. The time it takes to draw samples using NUTS also is dependent on $T$ with values of $T$ further from 1 generally taking longer. This is as expected as the target posterior for MCMC is exactly an isotropic Gaussian for $T = 1$ and becomes less Gaussian-like as $T$ varies from 1.

## C  Datasets and experimental setup

The details about the datasets used are outlined below.

**ImageNet [9]** Two downsampled versions at resolutions of $32 \times 32$ and $64 \times 64$ are used. The training set consists of 1.28 million images, and the validation set contains 50 000 images. The validation set is used as the test set, which is also done in [1, 6].

**LSUN [11]** This dataset contains multiple categories, but experiments are only performed on *bedroom*, *tower*, and *church* which are most commonly used in generative modelling papers. The training sets of bedroom, tower, and church, contain three million, 700 000, and 126 000 images, respectively. The validation sets of all three categories contain 300 images each. Image dimensions in LSUN are not constant, pre-processing steps from [1] are performed to downsample the smallest side to 96 pixels before taking $64 \times 64$ random crops. The validation set is used as the test set.

Figure S1: Coupling network architecture.

| Model/ Hyper-params. | Resolution | | | | | | | | | | |
|---|---|---|---|---|---|---|---|---|---|---|---|
| | 1 | 2 | 4 | 8 | 16 | 32 | 64 | 128 | 256 | 512 | 1024 |
| **ImageNet [9] 32** | | | | | | | | | | | |
| # Flow steps | 8 | 8 | 16 | 16 | 16 | 16 | - | - | - | - | - |
| Conv. channels | 128 | 128 | 128 | 128 | 128 | 256 | - | - | - | - | - |
| Patch size | 1 | 2 | 4 | 8 | 16 | 16 | - | - | - | - | - |
| Batch size | 64 | 64 | 64 | 64 | 64 | 64 | - | - | - | - | - |
| Parameter count | 4m | 4m | 8m | 8m | 8m | 32m | - | - | - | - | - |
| **ImageNet [9] 64** | | | | | | | | | | | |
| # Flow steps | 8 | 8 | 16 | 16 | 16 | 16 | 16 | - | - | - | - |
| Conv. channels | 128 | 128 | 128 | 128 | 128 | 256 | 256 | - | - | - | - |
| Patch size | 1 | 2 | 4 | 8 | 16 | 16 | 32 | - | - | - | - |
| Batch size | 64 | 64 | 64 | 64 | 64 | 64 | 48 | - | - | - | - |
| Parameter count | 4m | 4m | 8m | 8m | 8m | 32m | 32m | - | - | - | - |
| **LSUN [11] (bedroom/tower /church) 64** | | | | | | | | | | | |
| # Flow steps | 8 | 8 | 16 | 16 | 16 | 16 | 30 | - | - | - | - |
| Conv. channels | 16 | 64 | 64 | 64 | 64 | 128 | 320 | - | - | - | - |
| Patch size | 1 | 2 | 4 | 8 | 16 | 32 | 64 | - | - | - | - |
| Batch size | 64 | 64 | 64 | 64 | 64 | 64 | 64 | - | - | - | - |
| Parameter count | 68k | 1m | 2m | 2m | 2m | 8m | 93m | - | - | - | - |
| **CelebA-HQ [4]/ FFHQ [5] 1024** | | | | | | | | | | | |
| # Flow steps | 8 | 8 | 16 | 16 | 16 | 16 | 16 | 16 | 16 | 16 | 16 |
| Conv. channels | 64 | 64 | 64 | 128 | 128 | 128 | 128 | 128 | 128 | 128 | 128 |
| Patch size | 1 | 2 | 4 | 8 | 16 | 32 | 32 | 64 | 64 | 64 | 64 |
| Batch size | 64 | 64 | 64 | 64 | 64 | 64 | 64 | 64 | 64 | 64 | 64 |
| Parameter count | 1m | 1m | 2m | 8m | 8m | 8m | 8m | 8m | 8m | 8m | 8m |

Table S1: Hyper-parameters used with Wavelet Flow per level, and across the evaluation datasets.

**CelebA-HQ [4]** This dataset is an extension of the original CelebFaces Attributes dataset [8] which contains $1024 \times 1024$ images produced by applying inpainting and super resolution to aligned and cropped images. The 30 000 images in the dataset are separated into a training and test set using the same split as in [6]. The lower levels of the Wavelet Flow experienced overfitting, so a validation set is used to perform early stopping. The train, validation, and test sets contain 26 000, 1 000, and 3 000 images, respectively.

**FFHQ [5]** contains 70 000 high quality $1024 \times 1024$ aligned and cropped face images retrieved from Flickr. A training, validation, and test split were generated with 59 000, 4 000, and 7 000 images, respectively. As with CelebA-HQ lower levels of Wavelet Flow experienced overfitting and early stopping is used.

As stated in Sec. 3 of the manuscript, hyper-parameters are set differently for each dataset to keep the number of parameters below but within the range of compared methods. These hyper-parameters are shown in Table S1. The number of parameters for each variant is show in Table S2.

| Model | ImageNet [9] | | LSUN [11] | CelebA-HQ [4] & FFHQ [5] |
|---|---|---|---|---|
| | $32 \times 32$ | $64 \times 64$ | $64 \times 64$ | $1024 \times 1024$ |
| RealNVP [1] | 46m | 96m | 96m | - |
| Glow [6] | 66m | 111m | 111m | - |
| Wavelet Flow | 64m | 96m | 108m | 70m |

Table S2: Comparison of the total number of parameters used in Wavelet Flow, Glow, and RealNVP.

Figure S2: A Haar transformation on a single channel image. In practice, the Haar transformation is applied channel-wise. This transformation produces a low-pass and detail component denoted as $h_l$ and $h_d$, respectively. $h_d$ is comprised of the coefficients generated by the last three Haar filters that resemble vertical, horizontal, and diagonal derivative filters.

Coupling networks use a residual architecture [2], shown in Fig. S1. Inputs used to condition a coupling layer are concatenated to the input of its coupling network. As in [1], a *hyperbolic tangent* is applied to the scale component of the affine transform. The number of output channels in the convolutional layers is the same across the entire level of a Wavelet Flow.

Coupling layers are arranged into *steps*. Following Glow [6], the forward pass of a step is composed of an *activation normalization* (actnorm), *invertible* $1 \times 1$, and a coupling layer.

The Haar transformation used is shown in Fig. S2. The values of the low-pass component is effectively two times the box-downsampled image. In practice, it may be convenient to scale the low-pass component by $\frac{1}{2}$ to eliminate any inconvenient scaling factors.

## D  Additional results

Table S3 provides wall-clock times for training Wavelet Flow on each dataset, and at each level. Times also consider early stopping, where applicable.

Timing measured in seconds-per-image is provided in Table S4. Results are averaged over 100 iterations.

Additional quantitative results in bits-per-dimension for 5-bit CelebA-HQ at $256 \times 256$ is shown in Table S5.

Frechet Inception Distance [10] scores for Glow and Wavelet flow on LSUN $64 \times 64$ are shown in Table S6.

## E  Additional samples

Additional samples without annealing are shown in Figs. S3, S4, S5, S6, S7, S8, S9. Additional samples with annealing using the previously described MCMC method with $T = 0.97$, are shown in Figs. S10, S11, S12, S13, S14, S15, S16. Additional high resolution samples from CelebA-HQ, and FFHQ, are shown in Figs. S17, S18. Additional $8\times$ super resolution samples from CelebA-HQ, and FFHQ, are shown in Figs. S19, S20.

| Level | Resolution | FFHQ [5] | CelebA-HQ [4] | ImageNet [9] | LSUN [11] |
|---|---|---|---|---|---|
| 0 | $1 \times 1$ | 6 | 6 | 6 | 6 |
| 1 | $2 \times 2$ | 6 | 6 | 96 | 12 |
| 2 | $4 \times 4$ | 6 | 6 | 48 | 14 |
| 3 | $8 \times 8$ | 6 | 6 | 168 | 36 |
| 4 | $16 \times 16$ | 18 | 12 | 168 | 48 |
| 5 | $32 \times 32$ | 120 | 36 | 168 | 120 |
| 6 | $64 \times 64$ | 120 | 120 | 168 | 336 |
| 7 | $128 \times 128$ | 120 | 120 | - | - |
| 8 | $256 \times 256$ | 120 | 120 | - | - |
| 9 | $512 \times 512$ | 156 | 120 | - | - |
| 10 | $1024 \times 1024$ | 168 | 120 | - | - |
| Total | | 846 | 672 | 822 | 572 |

Table S3: Wavelet Flow training times in GPU hours on the evaluation datasets.

| Method | CelebA-HQ [4] $256 \times 256$ | ImageNet [9] $64 \times 64$ | LSUN [11] $64 \times 64$ |
|---|---|---|---|
| Glow [6] | 1.79 | 0.956 | 0.954 |
| Ours | 0.0147 | 0.0144 | 0.0293 |

Table S4: Training speed measured using seconds-per-image averaged over 100 iterations. Values for Glow were obtained by running their provided code on our hardware, a single NVIDIA TITAN X (Pascal) GPU, and without using any distributed computation frameworks. Note that the Wavelet Flow model for CelebA-HQ $256 \times 256$ is contained within the larger model for $1024 \times 1024$ images.

| Method | Celeba-HQ [4] $256 \times 256$ 5-bit |
|---|---|
| Glow [6] (Additive) | 1.03 |
| Ours (Additive) | 1.12 |
| Ours (Affine) | 0.94 |

Table S5: Quantitative results in bits-per-dimension on 5-bit CelebA-HQ $256 \times 256$.

| Method | LSUN [11] $64 \times 64$ | | |
|---|---|---|---|
| | Bedrooms | Tower | Church |
| Glow [6] (affine) | 60.03 | 54.17 | 59.35 |
| Ours | 121.20 | 87.20 | 93.08 |

Table S6: Frechet Inception Distance [10] scores on LSUN $64 \times 64$ between Glow (affine) and Wavelet Flow, both without annealing.

Figure S3: Samples from ImageNet ($32 \times 32$) without annealing.

Figure S4: Samples from ImageNet ($64 \times 64$) without annealing.

Figure S5: Samples from LSUN bedroom ($64 \times 64$) without annealing.

Figure S6: Samples from LSUN tower ($64 \times 64$) without annealing.

Figure S7: Samples from LSUN church ($64 \times 64$) without annealing.

Figure S8: Samples from CelebA-HQ ($256 \times 256$) without annealing.

Figure S9: Samples from FFHQ ($256 \times 256$) without annealing.

Figure S10: Annealed samples from ImageNet ($32 \times 32$), with $T = 0.97$.

Figure S11: Annealed samples from ImageNet ($64 \times 64$), with $T = 0.97$.

Figure S12: Annealed samples from LSUN bedroom ($64 \times 64$), with $T = 0.97$.

Figure S13: Annealed samples from LSUN tower ($64 \times 64$), with $T = 0.97$.

Figure S14: Annealed samples from LSUN church ($64 \times 64$), with $T = 0.97$.

Figure S15: Annealed samples from CelebA-HQ ($256 \times 256$), with $T = 0.97$.

Figure S16: Annealed samples from FFHQ ($256 \times 256$), with $T = 0.97$.

Figure S17: Annealed high resolution $(1024 \times 1024)$ samples from CelebA-HQ, with $T = 0.97$.

Figure S18: Annealed high resolution ($1024 \times 1024$) samples from FFHQ, with $T = 0.97$.

Figure S19: $8\times$ super resolution on CelebA-HQ. The smaller images (top) are the original images at ($128\times128$), while the larger images (bottom) are the annealed super resolution results ($1024\times1024$), with $T = 0.97$.

Figure S20: $8\times$ super resolution on FFHQ. The smaller images (top) are the original images at $(128\times128)$, while the larger images (bottom) are the annealed super resolution results $(1024\times1024)$, with $T = 0.97$.