[Reviews · NeurIPS 2020]

Review 1

Summary and Contributions: The paper proposes to use wavelets in normalizing flows. The authors expose the natural parallelism stemming from such wavelet decomposition to scale the training of this model to 1024x1024 images and enable faster training. The paper also uses MCMC algorithm (NUTS) to sample from an annealed distribution.

Strengths: The motivation and explanation of the technique is clear. The results are indeed competitive in terms of log-likelihood with accelerated training (claim of x15 acceleration). The paper shows an additional motivation for annealed sampling in terms of distribution matching.

Weaknesses: The global coherence of samples seems to be negatively impacted by that technique, which does not seem to be mentioned/addressed/studied in the paper. This is most visible in areas that should be sharp but are instead blurred by super-resolution. I personally suspect that this stems from the super-resolution stages being convolutional and not using global information. -- Authors' rebuttal acknowledged -- While the technique and initial experiments are interesting, a deeper study is lacking on the the point I've mentioned. I maintain my score.

Correctness: The claim and the methodology are correct.

Clarity: The paper is clearly written.

Relation to Prior Work: The prior work is discussed adequately and the paper explains how they differ from prior work. A relevant paper to cite would also be "Normalizing Flows with Multi-Scale Autoregressive Priors" by Shweta Mahaja, Apratim Bhattacharyya, Mario Fritz, Bernt Schiele, and Stefan Roth.

Reproducibility: Yes

Additional Feedback:


Review 2

Summary and Contributions: The paper introduces Wavelet flows, a normalising flow architecture for high resolution signals. It uses the wavelet transform to decomposes an input signal into its wavelet coefficients (I_0, D_0, D_1, …, D_r) and model them sequentially using separate conditional models that produce the detail coefficients conditioned on the previous resolution image. It thus simultaneously produces a model for all resolutions, and can also do resolution upscaling. They show that such an architecture is competitive with previous normalising flows like Glow for images on standard benchmarks while being faster to train, and also scales to high resolution. == Post rebuttal update == The authors in their rebuttal show convincing evidence for quantitatively better performance than previous methods. At the same time, the blurriness of the samples is not yet addressed, which is important for the high-resolution modelling to be useful, and architectural changes as suggested by Reviewer 1 or removal of the patch-wise training would be useful to ablate here to more clearly understand what causes the blurriness. I also agree with Reviewer 3's suggestion that exploring the choice of wavelets would be natural experiments to include. The quantitative results, especially the training speedup, seem strong enough though that I'd be ok to lean towards acceptance, and hence I revise my score from 5 to 6.

Strengths: The choice of the wavelet decomposition for use in normalising flows is novel, and they obtain a good super-resolution model for free. They also train quite efficiently compared to previous flows. For low temperature sampling, past work only successfully could use additive coupling blocks, but they show a way to get lower temperature samples from affine flow blocks too using MCMC. They obtain competitive performance in terms of logprob on ImageNet and LSUN benchmarks at lower compute, and have diverse samples on CelebA-HQ compared to Glow. The experiment studying effect of changing temperature to correctly match the entropy of detail coefficients is good.

Weaknesses: - While the aim is to use the wavelet transform to get high-resolution images, the samples (eg: Figure 2) qualitatively look quite blurry compared to the same 8-bit samples in past normalising flow papers like RealNVP/Glow/Flow++. It could be because they they used less compute, but the likelihood numbers are comparable on both Imagenet and LSUN and thus that suggests that it could be the wavelet decomposition isnt a good inductive bias for qualitative sample quality? Including some qualitative metrics like FID might help here. - It would be nice if the show comparisons to a baseline where they train a similar super-resolution model, but instead of the wavelet basis use our usual pixel basis, ie instead of producing the detail coefficients D_j from I_j, directly produce I_{j+1} from I_j. In this case, the decomposition is overcomplete since here H(I_n) = (I_0, I_1, …, I_n), but it would more clearly show if the quantitative performance benefits come from using the wavelet basis or from the super-resolution nature of the model. (Note this is equivalent to optimising log p(I_n) since p(I_{i-1}|I_i) is a delta function). This will also support the claims on Page 2 that modelling the residuals is better than directly producing the higher resolution image. (The paper seems to assume that overcomplete representations cant be used for flows, however since we’re actually optimising the discrete log prob with the dequantisation trick its ok to do the above decomposition)

Correctness: Yes, the methods are correct and and experiments are well described.

Clarity: Yes, paper is easy to understand and notation is clear.

Relation to Prior Work: Mostly. The super-resolution aspect should include comparisons to and discussion of previous flow approaches(eg SRFlow).

Reproducibility: Yes

Additional Feedback: Nits: 1. Patchwise training is a good idea, increases batch diversity and saves memory. However, some local features can depend on global information, is there a big enough receptive field? Example at high resolution, eye colors in a face. 2. Rearrange the tables to place them near where they’re referred to 3. Whats global variations in Line 108? Giving some intuition of what the wavelet coefficients mean could be useful 4. Not sure I understand what you mean by the multi-scale PixelRNN can’t be used for computing probability density of the high res images (Line 43). It can provide log p(x_r), and if you mean added the probability density of added content, can’t it just be log p(x_2r) - log p(x_r), where r is resolution?


Review 3

Summary and Contributions: This paper introduces a hierarchical structure for normalizing flows for density estimation and data generation based on wavelet transforms, allowing for a natural factorization of the data distribution based on different resolutions of the data. For density estimation, each image is fed into a sequence of wavelet transforms. Each wavelet transform takes an image and outputs a lower resolution image (obtained by a low-pass filter) and a tensor of detail coefficients (obtained by a high-pass filter). Repeatedly applying wavelet transforms to the output images leads to a set of detail coefficient tensors for each scale and a final 1x1x3 “image” representing the average intensity per channel. The original representation can be recovered from this representation with a sequence of inverse wavelet transforms. As a specific instantiation of a wavelet the authors consider a Haar wavelet which has a 2x2 filter size. Using the representations of the original image at different scales, the density of the original image is factorized into a product of distributions over the detail-coefficient tensors, each conditioned on the low-pass filtered image from the corresponding wavelet transform. The average intensity per channel is modeled with an unconditional density. Having obtained all of the low resolution images and detail coefficients from a sequence of wavelet transforms, each factor in the distribution is separately modeled with a (conditional) normalizing flow. This allows for training each of the flow components in parallel, as opposed to the common practice of training the flow levels in sequence in hierarchical flow architectures. Sampling does require sequential passes through each normalizing flow, starting from the coarsest representation of the image all the way up to the highest resolution, akin to super resolution. The authors claim that the parallel training of the different flow components can lead up to 15x faster training as compared to other hierarchical flows such as Glow. Furthermore, the proposed hierarchy allows the authors to train normalizing flows on images with a high resolution of 1024x1024 pixels. Finally, a significant part of the manuscript is devoted to sampling from an annealed density, as opposed to sampling from the density that the model was trained to optimize, as the authors argue that flows produce better samples when sampling from an annealed distribution. In order to sample with a temperature smaller than 1, the authors use MCMC to sample from the unnormalized annealed distribution. --------------- post-rebuttal update ------------------- Wall-clock time: I appreciate that the author’s rebuttal provided a more detailed comparison in terms of training times, this point has been addressed in a satisfactory manner for me. Choice of wavelets: the authors responded that learning a basis could be interesting for future work. As I stated in my review I would have liked to see some additional results that show the influence of the choice of wavelets (not necessarily learnable wavelets). One could imagine that this plays a role in the inductive bias of the model and the resulting sample quality (as reviewers 1 and 2 commented on the blurriness of samples). These seem to me like more natural experiments for a paper that introduces a wavelet-based hierarchy for normalising flows, rather than the extensive results on reduced temperature sampling. However, I am inclined to view this more as a different personal preference rather than a reason to reject a paper. With this in mind, I will retain my score of marginally above the acceptance threshold.

Strengths: 1. The proposed multi-resolution hierarchical structure naturally arises from the wavelet transforms and allows for parallel training of the flow components, which can in principle increase training speed. As normalizing flows are in general large models that take a long time to train, increasing the training speed while maintaining good performance is an important research direction to make flow models more practically useful. 2. The authors managed to scale their proposed method to high resolution data of 1024x1024 pixels, which is (to my knowledge) not done in previous normalizing flow papers.

Weaknesses: 1. As the paper’s main proposal is to use wavelet transforms to obtain a natural hierarchy of normalizing flows, it is surprising to me that the authors have not shown an exploration on the effect of the different possible choices for the wavelets. The influence of different wavelet choices on the sample quality and quantitative performance seems like an important topic to investigate, as wavelet transforms for normalizing flows form such a central part of the novelty of this paper. 2. One of the main claims of the authors is that the proposed method can be trained up to 15 times faster than other flow methods like Glow. Although it is likely that the parallel training of the flow components can decrease the training time, the comparison between training times presented in the paper has some issues. The authors only report the total number of GPU hours required to train their method to convergence on their hardware, and compare this to *estimates* of the total training time (in GPU hours) of Glow by looking at log files and through discussions on github. This seems problematic for several reasons. If the number of epochs used for training in the methods is not the same, then even if the estimates are correct, the numbers are hard to compare. A more informative comparison would be training time per epoch, or the time required for a single forward pass (with and without a parameter update). Since Glow’s code is publicly available, and the authors have used part of Glow in their own code, running such a comparison *on the same hardware* should not pose a problem, especially for the lower resolution images. Furthermore, the authors state that for the LSUN bedroom dataset, wavelet flows are 7.5 times faster than Glow, but wavelet flows also perform worse than Glow in terms of bpd. If you would train Glow to the same performance level as Wavelet flows, what would the difference in training time then be? Finally, the authors claim that Wavelet flows have a 15x speedup in training time for CelebA at a 256x256 resolution, but for this resolution of CelebA no results are shown in terms of bits-per-dimension for wavelet flows, so it is not clear if there is a trade off between training time and quantitative performance on this dataset. 3. A significant portion of the paper is devoted to sampling from the unnormalized annealed density with MCMC, with the authors arguing that samples from flow models are better at lower temperatures. As stated in the experiment section, the authors pick a temperature T=0.97 for the annealing temperature, which is very close to T=1 for which no MCMC is required and one can sample exactly from the flow model. This makes me wonder if running the MCMC procedure, which can also misbehave in high-D, is worth it when one can sample exactly at T=1.

Correctness: As mentioned above, although a training time speed up is likely to be present, the numbers provided in the paper to support this claim are hard to compare to those of other methods. Other claims and methods in the paper appear to be correct.

Clarity: Yes, the paper is clearly written.

Relation to Prior Work: In general the related work is sufficiently discussed. I do think a reference to SReC (https://arxiv.org/abs/2004.02872) and explanation of the difference would be in order.

Reproducibility: Yes

Additional Feedback: 1. The discussion on additive versus affine coupling layers and the annealing strength required for good samples is not the strongest part of the paper. The authors state that Figure 6 shows that the marginal density of the annealed additive model matches the ground truth marginal less well than that of the annealed affine model. However, even though the height of the peak seems better captured by the affine annealed model, the tails are better matched by the additive model. 2. In Fig 3c, is the panel with T=1 produced with mcmc sampling or with exact sampling from the normalizing flow model without MCMC? How do these two compare for T=1? 3. Where available, a comparison against better flow models such as Flow++ in table 1 would be appropriate, even if the authors mention that Flow++ uses more sophisticated dequantization which could be combined with the proposed method.


Review 4

Summary and Contributions: This paper proposed a method to train a multi scale Normalized Flow. Previously, Flow models are trained to learn a distribution of images on a single image size (e.g. 32x32). However, training on large images (e.g. 1024x1024) is very time consuming. This paper propose to firstly decompose a high resolution image into a pyramid of smaller images and learn one flow for each scale. A (predefined) Harr wavelet was used for decomposing an images into multiple scales. The experiments show that the proposed method achieve faster training (up to 15x) and on par bit per dimension. The authors also show that the proposed method can be applied to super resolution and sampling low resolution images.

Strengths: Learning the density function of high resolution function is a difficult problem. The existing method is normally slow in training. The proposed method provide a way to learn a density function of high resolution image in a fast way.

Weaknesses: While the result in table 1 on compare the proposed method with other baselines. The authors did not really compare them on HIGH RESOLUTION images. Results of both baselines and the proposed method are reported on images with 32x32, 64x64. On dataset with size 1024x1024, the authors only report the result of the proposed method since training on the baselines is umpractical. But, It would be nice to show some comparisons on dataset with bigger sizes such as 128x128 or 256x256. As we can see from table 1, with higher image size (I.e. 64x64), the proposed method shows worse result, it would be good to see whether this trend continue with increasing image size.

Correctness: the method is correct.

Clarity: the paper is well written and easy to understand

Relation to Prior Work: related work was clearly discussed

Reproducibility: No

Additional Feedback: After reading the feedback from the authors, I think my concerns are properly addressed. Although ablation studies of variant wavelet are not in the paper. The quality of high resolution images generated by the method is also not so good. This paper provide a novel method to train a flow network on high resolution images efficiently. Further studies such as different model architecture or hyperparameters may improve the quality of the generated images. I suggest to accept the paper.

[Author Response · NeurIPS 2020]

We thank the reviewers for their feedback and are glad that they found the paper to be clear, novel, and a well motivated direction of work. We will incorporate the answers/other feedback into the revised manuscript.

[**R1**, **R2**] **The general quality of samples seems to be negatively impacted.** We agree, the qualitative performance of Wavelet Flow (WF) appears to be somewhat worse than the Glow baseline but is still competitive in terms of bits-per-dimension. In particular, global coherence of fine details, eg, eye colour, gaze direction and hair texture, appear inconsistent over larger distances. It is not immediately clear exactly how the wavelet decomposition, receptive field, patch based training, and conditioning, interact to cause this. Initial experiments investigating these factors were inconclusive; however, we suspect that further architecture search, focused specifically on the conditioning networks, may help address these issues. Finally, we agree that quantitative metrics of image quality such as FID would be valuable. While this are not yet standard practice in normalizing flow (NF) papers which rely primarily on log-likelihood and qualitative assessment for evaluation, we will include an FID-based comparison in the final version.

[**R3**] **Exploration of different wavelet basis** We agree other wavelets could be potentially interesting. Indeed, we believe (see L290) that learning the basis should be possible and promising direction for future work; however as shown, the ability of any orthonormal wavelet basis to provide a natural, spatially coherent scale decomposition of a signal which fits naturally into the NF framework is, we believe, an important contribution in and of itself.

[**R2**] **Other baseline comparisons for super-resolution (SR).** SR is not claimed as our primary goal/contribution. Rather, it is a fortuitous byproduct of the conditional structure that WF enables. We included the SR results as a demonstration of this capability. A more thorough exploration of WF for SR is a promising direction for future work. We note that a comparison against a conditional structure $p(\mathbf{I}_i|\mathbf{I}_{i-1})$ is difficult because existing SR approaches of this form generate samples $\mathbf{I}_i$ from the full space $\mathbb{R}^{N \times N \times C}$, including regions of space which are inconsistent with the lower resolution image $\mathbf{I}_{i-1}$. In contrast, the wavelet construction means that SR sampling with WF produces images $\mathbf{I}_i$ which are *exactly* consistent with $\mathbf{I}_{i-1}$ in the sense that it is sampling from a lower dimensional manifold. Consequently, direct comparison of likelihood numbers is not possible. Finally, we note that, while training a NF with an overcomplete representation may be possible, it is not obvious how best to do so. Further, we believe that, because the wavelet transform is a bijection, it is a more natural fit with NFs, which are built on bijections.

[**R3**] **Further comparison of training time.** We recognise that our comparison of training time is imperfect. This was, in part, due to the challenges of training Glow with limited hardware resources. Using the public Glow code, we measured the average number of seconds-per-image over 100 iterations of training on our hardware. We provide timings for LSUN $64 \times 64$ (LS), ImageNet $64 \times 64$ (IN), and CelebA-HQ $64 \times 64$ (CA). Glow yields 0.954, 0.956, and 1.79 seconds-per-image on LS, IS, and CA, resp. WF yields 0.0147, 0.0144, and 0.0291 seconds-per-image on LS, IS, and CA, resp. Together, this is a speedup over over $60\times$. This is enabled, in part, because the smaller, simpler individual conditional models that makeup WF allow for larger batch sizes than is possible with a single monolithic Glow model.

[**R3**, **R4**] **Baseline Comparison on CelebA-HQ at $256 \times 256$.** Our experiments were focused on affine models on 8-bit data to avoid using separate models for quantitative and qualitative results, also motivated at L209. Note that, in our view, the practice of using a different model (additive vs affine) on different data (5bit vs 8bit) for qualitative vs quantitative evaluation is an unfortunate practice in the field which we sought to avoid. At the reviewers' request we provide quantitative results on CelebA-HQ $256 \times 256$ 5-bit to compare against the value reported in the supplemental material of Glow of 1.03 bpd. Using an additive and an affine WF model with the same setup and training time as the 8-bit model from the paper, we achieve 1.12 and 0.943 bpd, respectively. Finally, we note that higher resolutions do not produce worse results. Table 1 shows that on ImageNet at $64 \times 64$ Wavelet Flow performs better than the baselines. Further, the improvement over the baselines is even more significant than for ImageNet at $32 \times 32$.

[**R3**] **Value of MCMC with $T = 0.97$.** Yes, we believe that MCMC is worth it, even with $T = 0.97$. As shown in Fig. 3, there is a notable improvement in image quality from $T = 1$ to $T = 0.97$. See also Figs. S1-S15 in the Suppl. Mat. which contain more samples with $T = 0.97$ and $T = 1$ (i.e., without annealing). Note we believe the ultimate goal with NF is a model which produces high quality samples without annealing. MCMC allows us to determine how close we are to that goal and the fact that only a mild amount of annealing is required suggests that we are close.

[**R3**] **Marginal distribution of detail coefficients.** By definition, most detail coefficients are near the peak; capturing that region is likely to have the largest impact on image quality. Because of this we believe that fitting the peak is generally more important than the tails. However, instead of relying entirely on a qualitative comparison of the histograms, we have since computed the KL divergence between the generated histograms and the histogram of ground truth. In this case the annealed affine model (0.0309) matches better than the annealed additive model (0.0365).

[**R3**] **Difference between sampling from T=1 with and without MCMC?** For the case when $T = 1$ sampling with our MCMC method is effectively equivalent to exact sampling, albeit slower. Because we performed MCMC in the latent space, when $T = 1$ it becomes equivalent to performing MCMC on a Gaussian distribution with identity covariance. In this case, the MCMC method used (the HMC-based, No-U-Turn Sampler) is nearly exact.

[**R2**] **Intuition about the wavelet coefficients.** At $2 \times 2$, *global variations* capture whether the average intensity of the top/left halves of the image are brighter or darker than the bottom/right halves.

[**R4**] **Additional implementation details** Training details are in Sec. 3 and (cf L195) specifics of the architecture are in Sec. C of the supp. Code will be released with the final paper (cf L85) and is included with the submission.

[Meta-Review · NeurIPS 2020]

The paper proposed a normalizing-flow architecture based on a multi-scale decomposition obtained by a wavelet transformation. The benefits of the proposed architecture are faster training (due to improved parallelizability) and super-resolution. These are interesting and novel developments towards scaling up normalizing-flow models. The reviewers had a few concerns, some of which were addressed in the rebuttal (such as training time) while others remain (such as blurriness of samples), and pointed to several directions of improvement and future work. After the rebuttal, all reviewers recommend acceptance. I would strongly advise the authors to take the reviewers' feedback to heart when revising the paper. In particular, it would be good to add the improved comparison of training times to the revised version, and to discuss the issues regarding blurriness of samples, as well as discuss the possible improvements and directions of future exploration that the reviewers suggested.